# The Development of the New Process of Design for Six Sigma (DFSS) and Its Application

**Ching-Chow Yang [1], Yung-Tsan Jou [1,\*], Ming-Chang Lin [1], Riana Magdalena Silitonga [1,2] and Ronald Sukwadi [2,\*]**

1    Department of Industrial and Systems Engineering, Chung Yuan Christian University,
     Taoyuan City 320314, Taiwan; chinchow@cycu.edu.tw (C.-C.Y.); steven@motiontech.com.tw (M.-C.L.);
     riana.magdalena@atmajaya.ac.id (R.M.S.)
2    Department of Industrial Engineering, Atma Jaya Catholic University of Indonesia, Jakarta 12930, Indonesia
\*    Correspondence: ytjou@cycu.edu.tw (Y.-T.J.); ronald.sukwadi@atmajaya.ac.id (R.S.)

**Abstract:** The Six Sigma program has been widely adopted by industries worldwide since the late-1990s due to GE's successful implementation, though Motorola initiated it in 1987. Several quality experts, such as Juran and Feigenbaum and several researchers, have concluded that the poor quality in product design will cause vast and chronic costs, which the Six Sigma improvement may not eliminate. The focus of new product development is needed to transfer the quality improvement from the later phases of the Design Life Cycle to the early grades. It is necessary to adopt the "Design for Six Sigma" (DFSS) methodology to achieve the best performance in the early stages and focus on 'prevention' instead of 'problem-solving. In this research, we develop a new DFSS process, including Define, Identify, Measure, Design, Optimize, and Verify (DIMDOV), based on the in-depth study of DFSS and the author's rich consultant experience. The new DFSS process can be confirmed to be a robust methodology, used to assure the design quality of the product design and manufacturing process design by the application results.

**Keywords:** Design for Six Sigma; Design Life Cycle; Six Sigma; quality improvement

## 1. Introduction

Since 1950, Japan has learned and implemented the quality concepts, practices, and systems from the USA, especially the total quality control (TQC), which emphasizes quality assurance based on product life quality control starting from product design, manufacturing control, to after service. To perform TQC effectively, Japanese industries also direct tremendous effort in quality education and training for all employees, full participation and teamwork, continuous improvement, and the cultivation of quality culture. The unique quality system implemented in Japanese industries was called company-wide quality control (CWQC) [1].

During the 1970s and 1980s, Japanese industries possessed intensely global competitiveness and quickly gained a sizable market share in the western world, where it previously had limited experience in the effective implementation of CWQC. Japanese industries were overriding the worldwide markets by providing the customers with high-quality products at lower prices [2,3]. The solid Japanese competition caused American and western industries to benchmark Japanese CWQC practices and systems and learn from Japanese enterprises. Based on the reference to the characteristics of CWQC [1], western professionals in the area of quality management started to develop the system of Total Quality Management (TQM) [4,5].

TQM triggered the evolution of quality management, which has been widely adopted by industries and non-profit organizations worldwide. The development of TQM was also influenced by the western quality experts such as Deming, Juran, and Crosby [6], especially the Deming 14 points and Juran's quality trilogy [7,8]. TQM was thus an integrated model

of management philosophies, quality concepts, and practices. Since Kaizen (continuous improvement) is the core activity of CWQC, it is thus constant improvement that is the critical quality practice of TQM [9]. Several methodologies are used for Quality Improvement, for example, Quality Improvement Team (QIT), Quality Control Circle (QCC), and project management.

From the late-1990s onward, Six Sigma activity has become the robust improvement methodology adopted by most enterprises and non-profit organizations due to the very successful implementation of the Six Sigma program by General Electric (GE) in 1995 [10]. However, the Six Sigma program was initiated by Motorola in 1987. During the 1980s, Motorola executed the benchmarking of product quality from Japan in terms of the 'sigma' level. They observed that many Japanese products achieved a 'Six Sigma' level, but the average product quality level in the USA reached only 'Four Sigma.' The gap in product quality between Japan and USA stimulated Motorola to drive a five-year improvement program, named the "Six Sigma program". This program aimed to achieve the quality of their products to the Six Sigma level within five years [11,12]. Motorola implemented this program very successfully. The impressive results induced Allied Signal and General Electric (GE) to undertake a thorough implementation of the Six Sigma program in 1991 and 1995, respectively [10].

The Six Sigma improvement resulted in immense contributions, especially the financial results, as evidenced by Motorola, GE, and other organizations implementing the Six Sigma improvement. There are two kinds of implementation processes used in the Six Sigma method: DMAIC (Define, Measure, Analyze, Improve, Control) for Quality Improvement and DMADV (Define, Measure, Analyze, Design, Verify) for the design of the process; these processes are all used for product (including service), process, and system. Usually, the industries use DMAIC to improve the product quality but rarely use DMADV in the product design process. However, there are two issues related to the Six Sigma improvement that several researchers claim. The two issues are caused by some problems appearing in the design stage of a product, process, or system.

The first issue was considered very early by Juran; Juran's Quality Trilogy concluded that the deficiencies in the design stage would lead to substantial chronic waste in the operation process [8]. Since the poor cost is inherent in the process, the operating forces cannot offset the regular rubbish, even though they effectively used the Six Sigma improvement to counteract the original problems. Several researchers also have the same assertions. Yang and El-Haik [13] have asserted that the corrective actions used to improve designed entities' conceptual and operational vulnerabilities are costly and only marginally effective. Berryman [14] concludes that problems are discovered late in the product life cycle, and higher costs are needed to correct the issues.

The other issue is the very intricate problem in the design stage. It is unable to apply the Six Sigma improvement to eliminate the problem. It can be concluded that the Six Sigma improvement program insufficiently achieves the excellent Six Sigma quality level. Therefore, these two issues will induce the managers and engineers of the firms to focus on quality control in the design stage. The weakness of Six Sigma improvement and the costly correction of products in the late product life cycle causes the development of a methodology to focus on the stages of product design to attach to the Six Sigma quality level ultimately. Hence, this methodology is called 'Design for Six Sigma' (DFSS) [15–17]. It is acknowledged that the Six Sigma program is aimed at improving an existing process. Still, DFSS focuses on product design, especially in the early stages of the Product Life Cycle [16,17].

When the Six Sigma improvement cannot achieve the quality aim, it is necessary to return to applying the DFSS methodology [18], see Figure 1. DFSS will emphasize design thinking and preventative problem solving and intends to achieve breakthrough innovation. Usually, DMAIC focuses on statistical thinking and analytical problem solving and only results in incremental innovation.

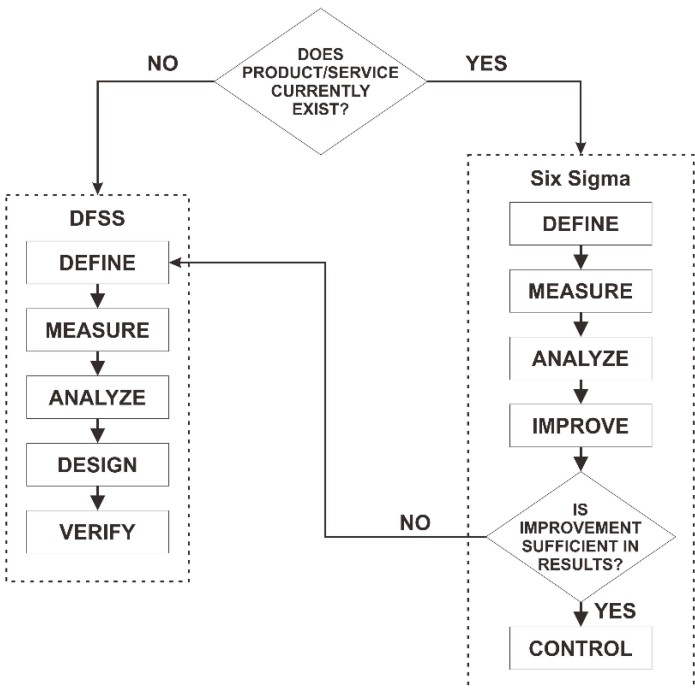

**Figure 1.** From DMAIC to DFSS.

The application of DFSS was coordinated by a collaborated team that can participate in the whole organization system to gain the proposed goals. The DFSS method has several problem-solving processes that can be used in various problems, such as Define, Measure, Analyze, Design, and Verify (DMADV), to improve the performance level of an extrusion process [13]; the DIDOV method (define, identify, design, optimize, verify) in designing a thermal design [19]; and IDOV (Identify, design, optimize, verify) in developing PIMCET [20]. Based on the previous research, it can be seen that it lacks a detailed analysis of the problem. Therefore, DIMDOV is used in this research to support the lack of previous research. This study aims to build and develop a new DFSS process, including Define, Identify, Measure, Design, Optimize, and Verify (DIMDOV), based on the in-depth study of DFSS and the author's rich consultant experience. The new DFSS process can be confirmed to be a robust methodology used to assure the design quality of the product design and manufacturing process design by the application results.

## 2. Materials and Methods

### 2.1. The Discussion of DFSS

Traditional new product development is an endless cycle of design–test–fix–retest. This product design will cause the firms to suffer from high development costs, longer delivery, and lower quality levels and may result in high failure opportunities for new product launches [13]. Therefore, the product designers intend to shift from problem-solving during the later phases of the design life cycle to problem prevention at the front-end phases, where product development is implemented by DFSS [13,14,17]. DFSS is a systematic engineering design process that can be characterized as an evolution from reactive design practices to predictive design performances, consisting of engineering design methods integrated with statistical tools and non-statistical techniques (for example, QFD, DOE, and TRIZ). The aim is to allow prediction and improvement in quality performances before manufacturing [14].

DFSS is performed to translate customer needs and expectations into design requirements and product functions, probe and adopt effective design alternatives, verify that the design is robust, and achieve the DFX (X will be Six Sigma, costs, or manufacturing) targets [21,22]. Several researchers assert that the effective implementation of DFSS will result in several benefits [14,15,17,18,23,24] as follows:

- The customers' needs and expectations are realized;
- Most of the problems are prevented early in the Design Life Cycle;
- The product design is robust;
- Very few problems are released from the design stages to the manufacturing process;
- Reduce the number of design changes and iterated tests;
- Reduce the product development time;
- Reduce the cycle time of product production;
- Reduce time to market for new products;
- Reduce the total costs of new product development;
- Enhance the quality and reliability of the products;
- Reduce warranty costs and after-service costs;

The effectiveness of the DFSS application is influenced by the implementation process and steps, as well as the usage of tools and techniques [25]. However, the actual connection of DFSS with the critical contents of engineering design practices has been less emphasized in the literature [15]. The Six Sigma improvement process DMAIC is well-understood and commonly used by the industries. We will discuss it in the next section. Although several researchers use DMADV as the DFSS process, it is unsuitable for the new product design. Since the users do not commonly accept DMADV, many different types of DFSS processes are available in the literature [14,15,24]. We list several different DFSS processes as follows:

- DMADV [22,24,26,27];
- ICOV (Identify, Characterize, Optimize, Verify) [13,28];
- ICOV (Identify, Characterize, Optimize, Validate) [29];
- IDOV (Identify, Design, Optimize, Verify) [24];
- IDOV (Identify, Design, Optimize, Validate) [23,30,31];
- IDOV (Identify, Develop, Optimize, Verify) [22];
- IDDOV (Identify, Define, Develop, Optimize, Verify) [22];
- DCOV (Define, Characterize, Optimize, Verify) [24,32];
- CDOV (Concept, Design, Optimize, Verify) [18,24];
- DCCDI (Define, Customer, Concept, Design, Implement) [20];
- RDIDOV (Recognize, Define, Identify, Design, Optimize, Validate) [24].

Additionally, very few other DFSS processes that Shahin [24] considered are absent in the above list, since industries seldom use them.

As we thoroughly discuss these DFSS processes, we will find out about some shortages in operation. Thus, we will develop an ideal DFSS process in the next section, which is applied by the industries, based on the above processes' analyses and the authors' industrial consultant experience.

### 2.2. The Development of a New DFSS Process

The implementation of the DFSS process is expected to achieve the following results:

- Embedding the customer's needs and expectations into product functions and standards;
- Optimized parameters design, which can be controlled well and attached to Six Sigma level;
- To be a robust design that will be insensitive to variations in the noise factors;
- Easy release from the design stage to the manufacturing process with minimal problems;
- Most of the issues in the manufacturing process can be predicted and prevented in the early design stages.

These results partially depend on correctly determining key inputs and process factors based on the Inputs–Process–Outputs analytic model, see Figure 2. The output results, $Y$ ($Y$ may be a matrix ($Y_1$, $Y_2$, ... $Y_1$)), is dependent on the key inputs, and critical factors in the process [15] that is,

$$Y = f(I_1, I_2, \ldots . I_m; \ X_1, \ X_2, \ldots \ X_n) \tag{1}$$

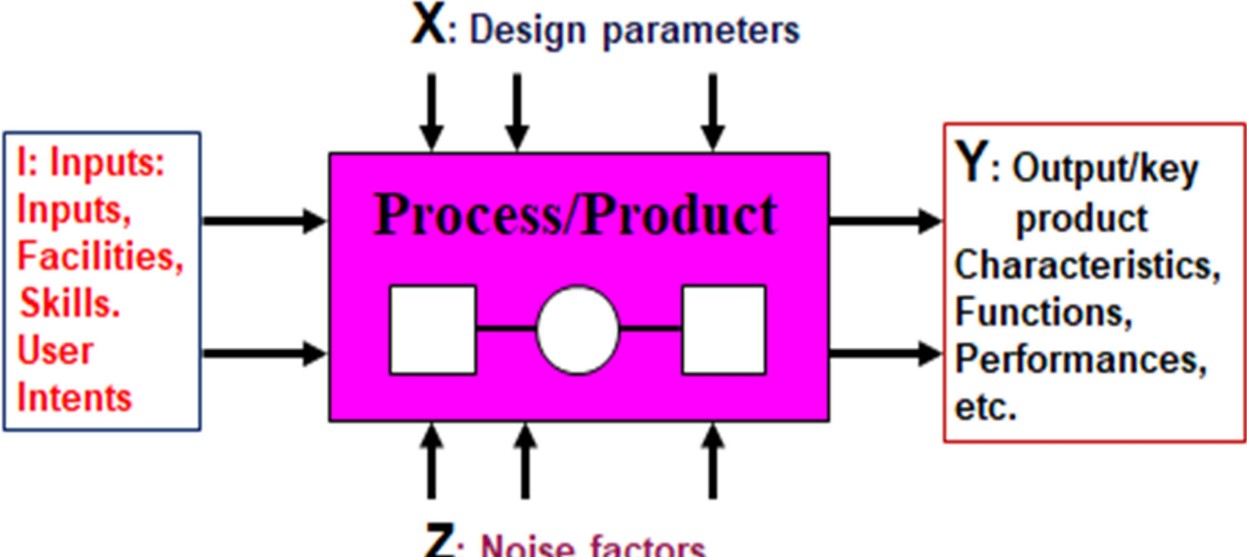

**Figure 2.** The I-P-O analytic model.

$I_1, I_2, \ldots I_m$ are the key inputs, and $X_1, X_2, \ldots X_n$ are the critical factors in the process. It is preferable that we can discover an axiomatic relationship for Equation (1). These key input factors in the process are needed to be under control. To find out their optimal control conditions or optimal parameter design, which can be well controlled, is a prerequisite to measure these key inputs and critical factors before designing the optimal conditions and parameters. It can be concluded that before the step 'measure', identifying critical vital factors in the process and the axiomatic equations with output results $Y$ is essential. Thus, we need a step 'identify' before the stage 'measure', which is neglected by the process 'DMADV'. This is why several methods of DFSS mentioned above have the step 'identify'.

Based on the above discussion, the referred DFSS processes, and the author's consultant experience in product development, we developed a new DFSS process as Define, Identify, Measure, Design, Optimize, and Verify (DIMDOV), see Figure 3. Since the implementation items and the usage of management tools are very critical to the implementation results, in the following, we will describe the operation items and the use of tools in each step of the DFSS process.

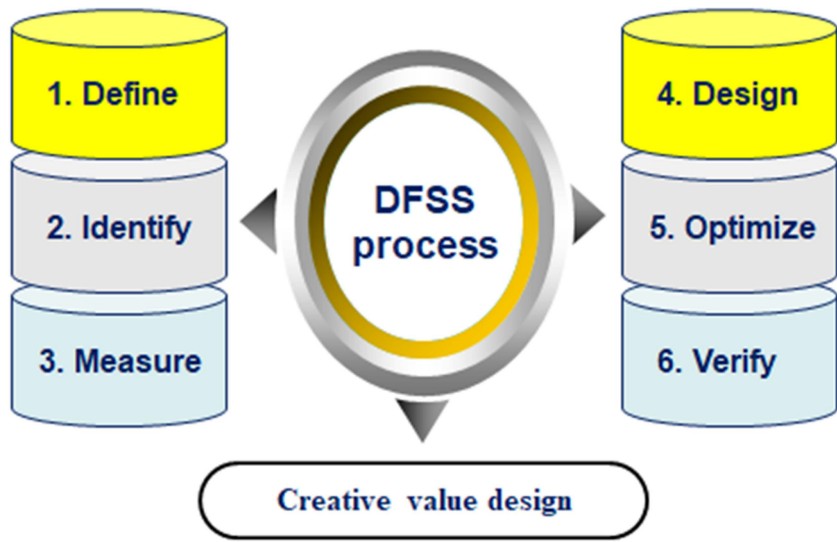

**Figure 3.** The conceptual framework of the new DFSS process.

*2.3. Research Methodology*

In this section, we will describe the implementation of each step of DFSS.

*The 'Define' step*

Usually, any new product development will be performed by using project management. Still, before the project is decided, it is needed to discuss the issues of customer requirements, market analysis, costs, profits, etc. As the new product is worth developing, we must initiate the project first, create the team charter, and set the project objectives. Thus, in the 'define' step, we can consider three sub-steps involving work items (see Figure 4).

(1)  Initiate the project

- Understand the customer's needs;
- Capture the customers' latent needs (Questionnaire customer, complaint);
- Perform the analyses of product development;
- Decide the development project.

(2)  Develop team charter

- Organize the project team;
- Determine the scope of the project;
- Determine the roles of team members and the work's assignment;
- Plan the schedule and budget;
- Schedule the project review.

(3)  Establish project objectives

- Set up the project objectives;
- Determine the milestones of the project;
- Use key performance indicators to evaluate the project;
- Collect baseline data.

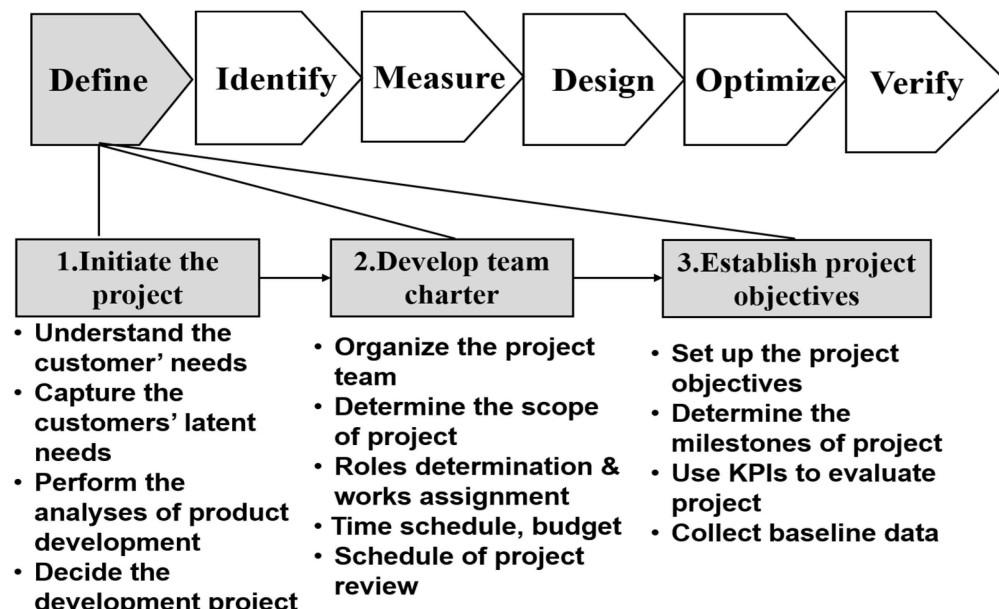

**Figure 4.** The 'Define' step of DFSS.

Several management tools can be used in this step, such as market research, benchmarking, Voice of Customer (VOC), affinity diagram, QFD (Quality Function Deployment), project management, and risk management.

*The 'Identify' step*

This step is crucial because many critical issues need to be determined. We also consider three sub-steps and their involved work items of the 'Identify' action (see Figure 5),

as follows. We first define the Critical Qualities (CTQs) by analyzing the customer needs, especially the latent needs, and the critical outputs related to customer needs. Then we set up the necessary functions and the specification of CTQs. Since the vital qualities are realized in the manufacturing process, we need to find out the Critical to Processes (CTPs) and the related key input variables and output variables. Additionally, the realization of CTQs' needs to develop manufacturing techniques and innovative alternatives.

(4)  Determine the real CTQs

- Prioritize the customer requirements;
- Determine the critical qualities (CTQs), *Ys*;
- Set up the specification of CTQs;
- Determine the functional specification.

(5)  Develop CTPs from CTQs

- The ability to realize CTQs and the essential process outputs, KPIs;
- Determine the critical processes (CTPs);
- Analyze the key factors of the process, *Xs*;
- Analyze the critical factors of materials and inputs.

(6)  Analyses of techniques

- Process analysis by using concurrent engineering;
- Set up the milestones of process techniques;
- Create the innovative alternatives;
- The development of new techniques.

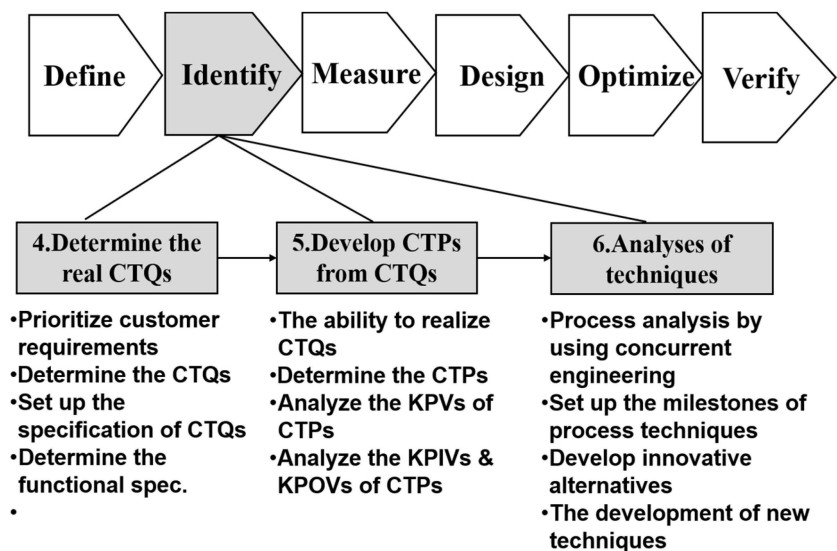

**Figure 5.** The 'Identify' step of DFSS.

This step can use many tools to enhance the implementation results, for example, QFD, Pugh selection matrix, force field analysis, SIPOC (Supplier-Input-Process-Output-Customer), Kano model, refined Kano model [6], and the Analytic Hierarchy Process.

*The 'Measure' step*

With the exception of DMADV, most of the DFSS processes mentioned above neglect the 'Measure' step. We think measurement is critical for the optimal design of key factors *Xs* and the development of targets of Ys and KPIs. All the *Ys*, KPIs, and *Xs* are needed to be measured; we need to develop the measurement methodologies and machines, and even the system for these critical factors and CTQs. We also need to set up all the metrics with the SMART (Specific–Measurable–Achievable–Relevant–Timed) rule and assure the accuracy of the measurement instrument. It is better to develop the axiomatic theory *Ys = f (Xs, Zs)* (where *Zs* are the voice factors) to control the key factors *Xs* and the environmental

factors *Zs*. The three sub-steps of the 'Measure' step and their involving work items are considered as follows (see Figure 6).

(7)　Measure the key factors

- Measure all the KPIs *Ys*;
- Develop the measurement method of *Xs*;
- Develop the measurement method of critical factors of inputs;
- Analyze the relation between *Ys* and *Xs*.

(8)　Measurement system

- Establish the measurement machines and methodologies;
- Build the measurement system of *Ys* and *Xs*;
- Construct the spec. of *Xs*;
- Create the axiomatic theory *Ys = f (Xs, Zs)*.

(9)　Assure SMART metrics

- Measurement Systems Analysis (MSA);
- Set up the targets of Ys with SMART;
- Develop the control way of the specification of *Xs*;
- Assure the accuracy of the measurement instrument.

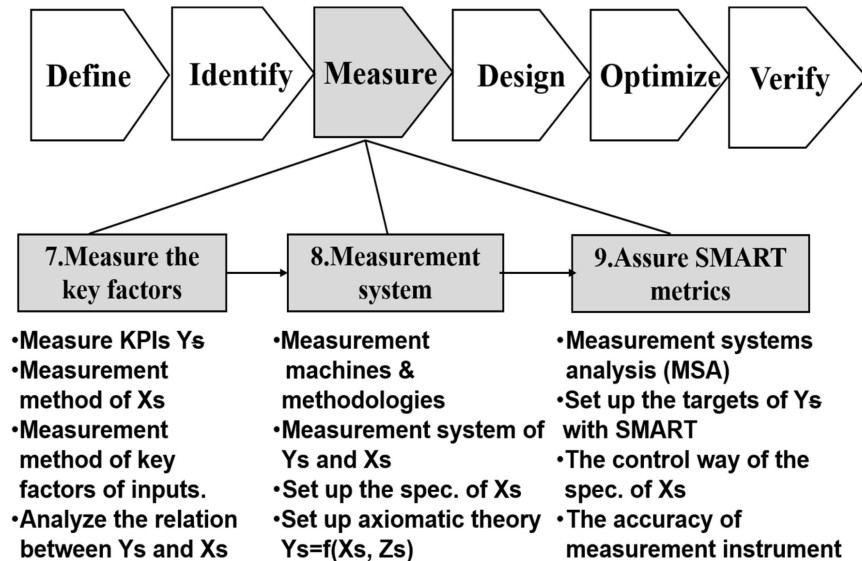

**Figure 6.** The 'Measure' step of DFSS.

In this 'Measure' step, the following management tools can be used to effectively perform the work items, evaluation matrix, prioritization analysis, MSA, AHP, internal metrics, objective management, and Objectives and Key Results (OKR).

*Axiomatic Design*

Axiomatic design is a design method that establishes the essential information and basic design elements. In these unique circumstances, a logical hypothesis contains principal information regions as discernments and understandings of different substances and the relationship among those primary areas. Those primary areas are related to mathematical expressions and categorizations of phenomena or objects, models, etc., and are more abstract than real-world data observation [33].

Axiomatic design is a scientific and structured framework applied to figure out the design process. It works based on mathematical and analytical thinking and tools with trials and error experiments [34].

The design model of this approach contains four domains: the customer domain, the functional domain, the physical domain, and the process domain. Those domains are

analyzed as "what we have to achieve in designing a product?" and "how to satisfy the need of the product's user" [35].

Figures 7 and 8 show the relationship between each domain in axiomatic design approaches. The design process goes from the left side of the part to the right side of the field. The designer can go back to the left side according to the previous ideas.

1.  Customer domain: This domain identifies Customer Attributes (*CA*) and focuses on what the customer needs and desires in the product. The customer domain describes the customer preference about the product and things to avoid in the development, system, and process;

2.  Functional domain: This domain identifies the Functional Requirement (*FR*), which describes what the design must do. *FR* is developed to satisfy the customer attributes and features;

3.  Physical domain: This domain is characterized by the Design Parameter (*DP*). It describes the act of a physical component to satisfy the functional requirement of the product to fulfill the design process. This domain applies the conceptualized product already developed in the applicable requirement domain. *DP* describes what the design looks like, so the physical item of the product is already fulfilled *FR* independently;

4.  Process domain: it figures out the necessary process of the physical domain in the manufacturing process that determines the product design. Process Variables (*PV*) explain how the *DP* is produced. *PV* can be a manufacturing process, such as machining, injection molding, and assembly. *PV* has to be developed one by one with the *DP* independently.

In identifying the relationship between one domain and others, the axiomatic design uses a matrix to see whether each domain relates or not.

$$[FR] = \lfloor A \rfloor \lfloor DP \rfloor \tag{2}$$

Equation (2) shows that the *A* identifies the product design that develops between *FR* and *DP*. It is typical for the element of $A_{ij}$ to be represented with an *X* if there is a strong relationship between *FR* and *DP*. Otherwise, 0 is used to express there is an insignificant (or no) relationship. Moreover, the process design matrix is characterized as *B* in the below Equation (3).

$$[DP] = \lfloor B \rfloor \lfloor PV \rfloor \tag{3}$$

According to the type of design matrix (*A*). There are three types of the design matrix, which related to the robustness of the product

$$\begin{bmatrix} FR_1 \\ FR_2 \\ FR_3 \end{bmatrix} = \begin{bmatrix} X & 0 & 0 \\ 0 & X & 0 \\ 0 & 0 & X \end{bmatrix} \begin{bmatrix} DP_1 \\ DP_2 \\ DP_3 \end{bmatrix} \tag{4}$$

The matrix in Equation (4) shows the diagonal design matrix, which identifies as an uncoupled matrix. This matrix is a powerful design concept because each *FR* fulfills one *DP* only. Hence, it does not need any decomposition process in this design matrix.

$$\begin{bmatrix} FR_1 \\ FR_2 \\ FR_3 \end{bmatrix} = \begin{bmatrix} X & 0 & 0 \\ X & X & 0 \\ X & X & X \end{bmatrix} \begin{bmatrix} DP_1 \\ DP_2 \\ DP_3 \end{bmatrix} \tag{5}$$

The triangular matrix (5) shows that the design can be accepted and redundant due to the number of *DP* larger than *FR*, which does not affect the independence axiom. Hence, decomposition is not necessary to do.

$$\begin{bmatrix} FR_1 \\ FR_2 \\ FR_3 \end{bmatrix} = \begin{bmatrix} X & X & X \\ X & X & X \\ X & X & X \end{bmatrix} \begin{bmatrix} DP_1 \\ DP_2 \\ DP_3 \end{bmatrix} \tag{6}$$

The coupled matrix is shown in Equation (6). This kind of matrix defines that the design needs further decomposition until it is robust. The entire design matrix identifies the coupled matrix. Each *FR* is related to another *FR* due to the number of *DP* being less than *FR*. Therefore, decomposition is necessary.

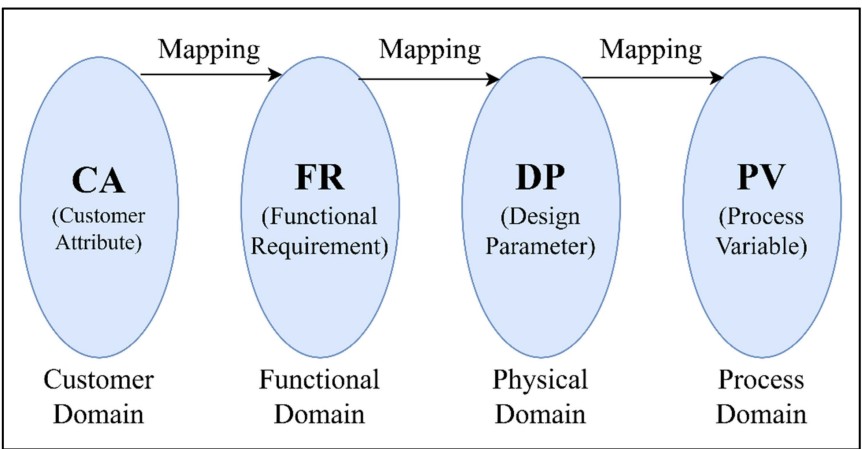

**Figure 7.** Axiomatic design domain.

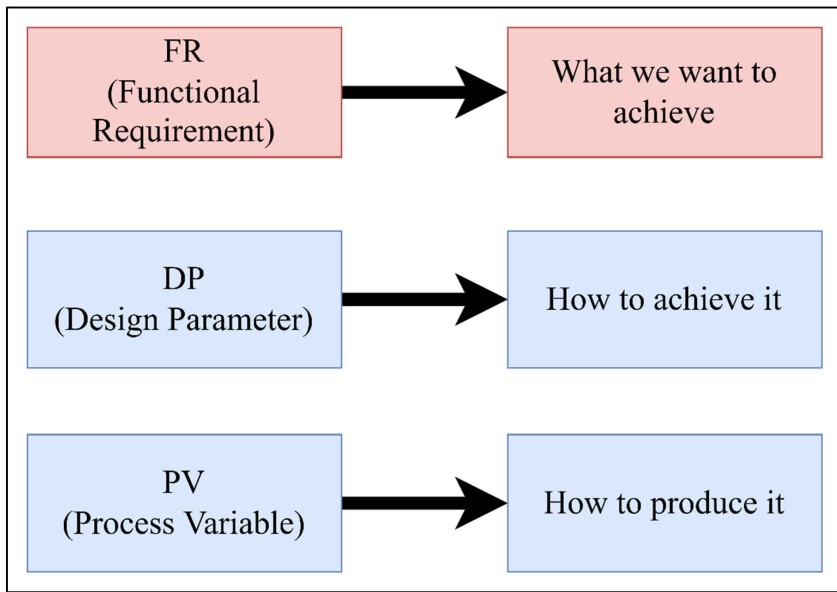

**Figure 8.** The meaning of different variables according to the domains.

*The 'Design' step*

The 'Design' step is the foremost step of the DFSS process. The design progress will be feasible, based on the actions and implementation of the previous actions. To fulfill the main objectives of DFSS, the focus of design steps is on how to reduce the problems in the later stage of the product life cycle, especially in the manufacturing stage. Therefore, design for X (X may be manufacturing, quality, costs, reliability, etc.) is the centric action of product design. Furthermore, upon building the 'Design Failure Mode and Effects Analysis', set materials and input conditions standards increase the optimal target and control tolerances of key factors *Xs*; control of the variation of *Xs* are also essential in the design step. By

convention, we divide the 'Design' step into three sub-steps and list involving work items as follows (see Figure 9).

(10)  Formulate concept design

- Select the best alternatives;
- Develop DFMEA;
- Confirm the design feasibility of CTPs;
- Design analysis for crucial process steps.

(11)  Determine the needed DFX

- Carry out DFM design;
- Material selection and design change according to DFC analysis;
- Carry out DFR analysis;
- Conduct DFQ analysis based on the CTQ targets.

(12)  Establish target and tolerances

- Establish target and tolerances for key process factors *Xs*;
- Analyze effects of variation;
- Empirical and analytic design;
- Assess process performance and process capability analysis.

The management tools that can be used in this 'Design' step are the following: QFD, DFX, SIPOC, process analysis, DFMEA, simulation, and TRIZ.

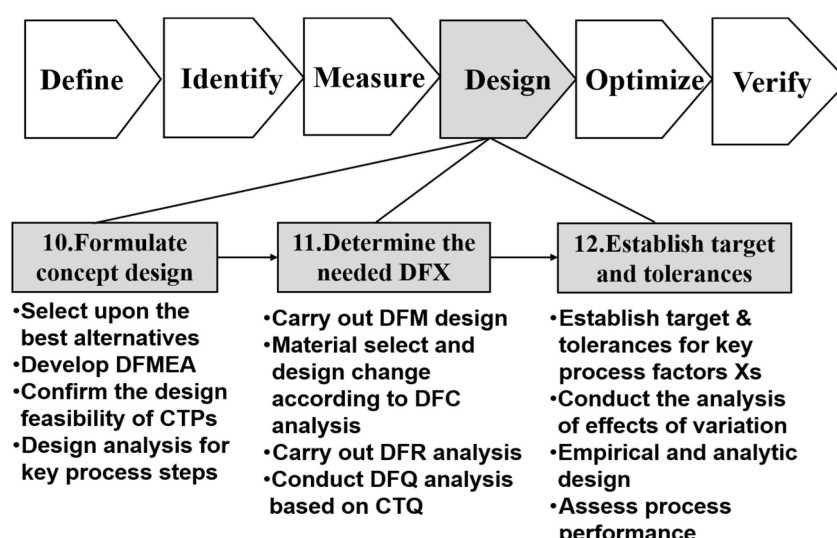

**Figure 9.** The 'Design' step of DFSS.

*The 'Optimize' step*

The objective of this 'Optimize' step is to achieve a robust design and optimal control of the manufacturing process. First, we need to realize the DFMEA to reduce the problems during the design phase and in the later manufacturing process. Then, we will focus on optimizing the parameters of key factors in the process, especially the manufacturing machines and the materials and input conditions standards. In addition, the skill training for employees, the confirmation of the precise capability of the machines, and how to control the optimal parameters are also critical works in this 'Optimal' step. As a result, we expect that the targets of the output quality and CTQs can be achieved. The three sub-steps of the 'Optimize' stage and the actions included in these subsections are considered as follows (see Figure 10).

(13)  Develop a robust design

- Perform the DFMEA;

- Create the methods to be robust;
- Apply DOE to find out optimal production parameters;
- Set up the precise 'Standard Operating Procedures (SOPs).

(14) Evaluate the manufacturing engineering

- Confirm the capability of manufacturing machines;
- Set up the precise parameters for manufacturing machines;
- Plan the skill training for employees;
- Set up the standards for materials and input conditions.

(15) Assurance of critical quality

- Ensure the process capability for key steps;
- Assure the quality performance for CTQs;
- Well control the quality of the output in the whole process;
- Establish the axiomatic formula for equipment parameters.

We also list the management tools in this 'Optimize' step: Axiomatic theory, DFMEA, DOE, Taguchi method, process capability analysis, simulation, and reliability analysis.

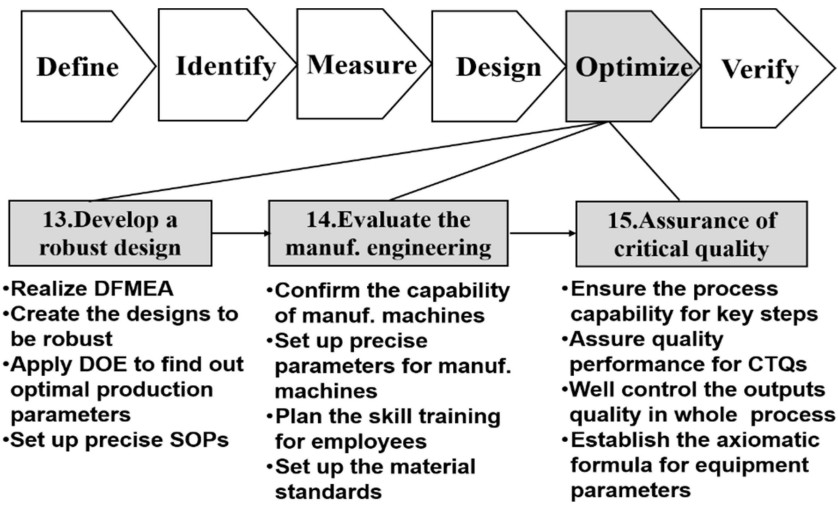

**Figure 10.** The 'Optimize' step of DFSS.

*The 'Verify' step*

This is the final step; the main works in this step are to approve the optimal design works in the previous steps. The pilot in the production line realizes the agreed actions. It is possible some design changes and correct parameters are needed to control the targets and tolerances for the key factors *Xs*. Then we must evaluate the quality and reliability that certainly achieve customer needs and expectations, and then pass the product certification by the customer first. We need to adopt preventive actions to avoid any unexpected problems. In the final, we set up the Statistical Process Control (SPC) system and 'The Process Failure Mode and Effects Analysis' system and effectively implemented these systems. The three sub-steps of the final 'Verify' step and their involving action items are demonstrated in the following (see Figure 11).

(16) Demonstrate process capability

- Small batch pilot run in the production line;
- Confirm the process capability of critical factors;
- Necessary design changes and parameters correct;
- Control the target and tolerances for key factors *Xs*.

(17) Evaluation by customers

- Maintain the excellent performance of CTQs;

- Assure the product reliability;
- Achieve the customer needs and expectation;
- Pass product certification for the first time.

(18) Set up a quality management system

- Set up and implement the SPC system;
- Build the leading control indicators in the process;
- Set up and enforce the PFMEA system;
- Effectively use the preventive actions.

The management tools often used in this 'Verify' step are PFMEA, SQC, Poka-Yoke method, capability analysis, prevention, and customer evaluation.

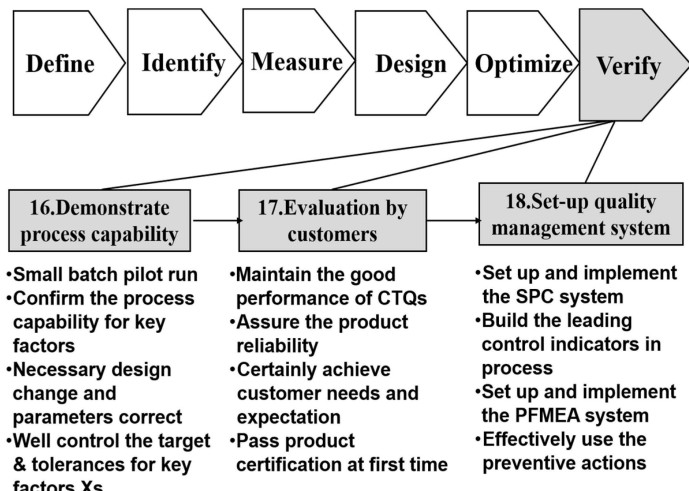

**Figure 11.** The 'Verify' step of DFSS.

## 3. Results

Currently, the product's quality is a primary consideration for the company. It is caused by the loyalty of customers, which is equal to the product's reliability. Therefore, to ensure the quality of the product meets customer expectations, companies need to carry out further product analysis to minimize defects and rejections. Motion Technology Electric and Machinery, Co., Ltd. (MTM), Taoyuan City, Taiwan (R.O.C.), is being authored as the subject limitation of the case study. This company has been one of the most well-known companies in Taiwan since 2002, focusing on brushed and brushless DC motors. It has also been devoted to the design and production of high-precision engines for use in medical appliances, electric wheelchairs, mobility scooters, and all kinds of elevators and transmission systems.

Several high-technology companies in Taiwan (such as Unimicron Corporation and Motion Technology Electric and Machinery, Co., Ltd.), have been implementing the new process of DFSS. The present study took Motion Technology Electric and Machinery, Co., Ltd. (MTM) as the subject for a case study.

The company's leading Research and Development (R&D) facility and operational headquarters are in Taoyuan County, Taiwan. The base has approximately 170 skilled employees, more than 60% of whom are 'cutting-edge' technology researchers. Many are responsible for product R&D, sales and marketing, and logistical management. The core of the company's manufacturing is in brushed and brushless DC motors, devoted to designing and producing high-precision engines for medical appliances, electric wheelchairs, mobility scooters, and all kinds of elevators and transmissions systems.

The company also boasts expertise in transmission system design and integration, as well as research and development of gear wheel and drive mechanisms. It has provided Original Equipment Manufacturer (OEM) and Original Design Manufacturer (ODM)

services to international industry leaders and developed products that comply with international requirements.

Over the years, MTM has significantly increased its investment in core technology development. In 2020, the acquisition was NT$12,680,000, and in the year 2021 rose to NT$53,680,000.

At the time of this study, MTM had completed three Six Sigma projects. Information was obtained through interviews with quality improvement leaders and members. Based on the interview, the company obtained the 5.0, 5.1, and 5.1 sigma levels in 2020. The sigma level was calculated for the defective product of model SCC4M2436A2CR6BA. The list of interviewees for MTM is shown in Table 1. Based on the interviews and documentary evidence, we analyze the implementation process of the new technique of Design for Six Sigma projects and discuss the findings.

**Table 1.** Interview data.

| Organization | Interviewee | Designation |
|:---:|:---:|:---:|
| MTM | 1 | Chief Executive Officer |
| MTM | 2 | Head Department of Quality Control |
| MTM | 3 | Head Department of Research and Development |

According to the interviewee data, several main problems made the product receive a rejection by the customer due to the Not-Good (NG) product. The interviewees are chosen based on their expertise and knowledge regarding the MTM's production. Their experiences at MTM proved that they were capable to be interviewees to support this research and to ensure the effectiveness of their scoring. To overcome the problem, the DFSS approach is necessary to solve the problem and improve the quality of the product through Design, Identify, Measure, Design, Opti-mize and Verify Process (DIMDOV).

### 3.1. Define

For several years, MTM received customer complaints from companies in the Netherlands about the motor's defective product (refer to Figure 12). The protests were becoming severe due to the improper function of the product (Table 2). Therefore, this step focuses on defining the product problem that needs to be considered more as the critical problem to solve.

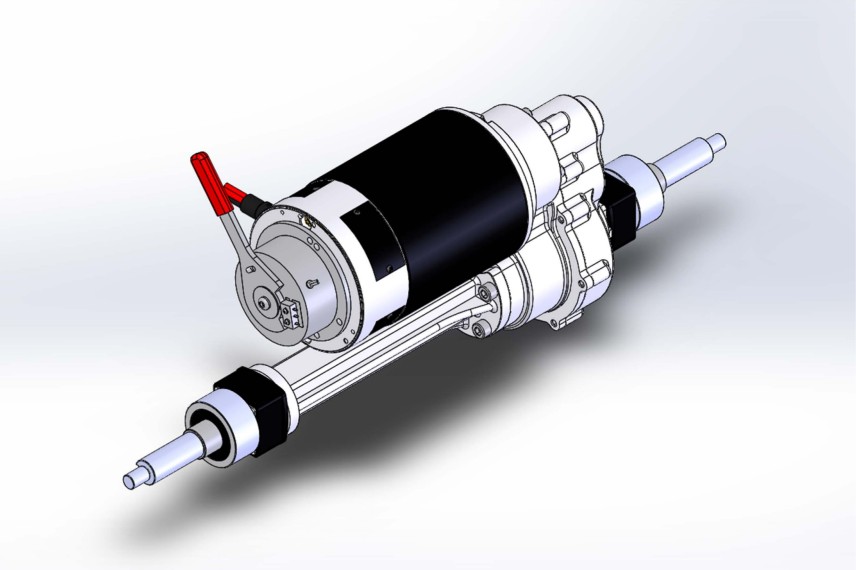

**Figure 12.** Motor model SCC4M2436A2CR6BA.

**Table 2.** Customer Need Identification.

| No | Model | Customer | Complaint | Customer's Needs |
|----|-------|----------|-----------|------------------|
| 1 | SCC4M2436A2CR6BA | Sunrise company (Netherlands) | Improper placement position of micro switch. The protrusive condition of the microswitch | The micro switch is well installed. The switch button is in the correct position. |

Figure 13a shows the defective brake that was taken from the customer complaints. The red circle in Figure 13a shows that the main problem is caused by the improper installment of microswitch. The microswitch is not installed correctly in the correct position, so the brake did not work functionally. In addition, The red circle in Figure 13b exposes that part of the microswitch is detached from its main component. This is the leading cause of the microswitch not being installed perfectly.

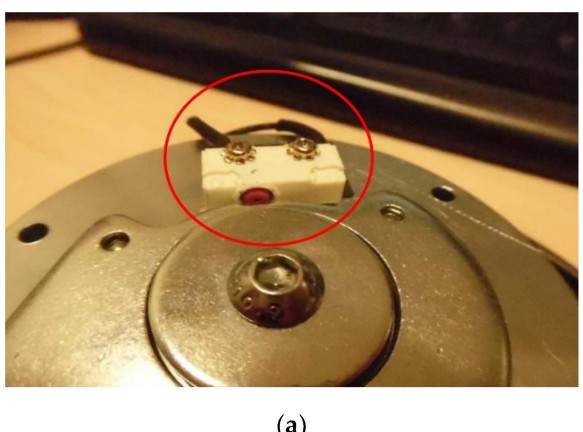

(**a**)

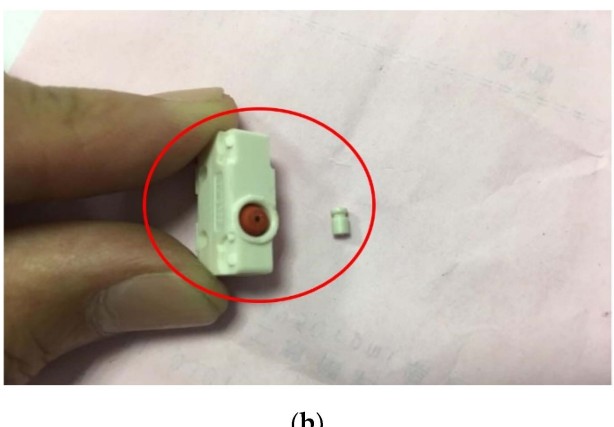

(**b**)

**Figure 13.** Defective brake from customer complaints with a different point of view (**a**,**b**).

Table 2 describes the customer need for the motor model SCC4M2436A2CR6BA. The customer need for this model is conducted based on the interview result (Appendix A).

*3.2. Identify*

The quality control of MTM already carried out some tests for 512 products, based on the company's stock. According to the results, four products are Not-Good (NG). It is proved that the related product needs to be analyzed further to minimize the defective product in the future. The result of the checking test can be seen in Figure 14. The main problem with faulty brakes is due to an improper placement position and protrusive condition of the micro switch. These problems have the same percentage at 0.4%. Hence, the same treatment solution must be applied in the next stage. Although, the rate of defective brakes is lower than that of good brakes. It is still necessary to be analyzed and improved. The company must be aware of the defective product through the product design and development process to minimize the NG product. The more NG products are created, the less customer trust will be obtained.

According to the customer complaint in the previous stage, the company needs to identify the problem in more detail. Therefore, the solution can be obtained based on the customer's needs. To determine the problem, the authors use the Critical to Quality (CTQ) tree to emphasize the root of customer needs.

The CTQ (Figure 15) is created based on what customers require, not the company's needs. Hence, this tree helps the user to solve customer complaints significantly. The high-quality product is the paramount latent need of MTM's customers, related to their feedback. Common CTQs involve time (cycle time, service time, and turnaround time), time to respond and restore customer complaints, and providing accurate and timely

information to customers. In some NG products, the users acess some information that needs more attention, to be analyzed. Proper installation of micro switch and normal switch button position is a significant consideration from the customer for the brake. These considerations are the main problems that make the brake not work well.

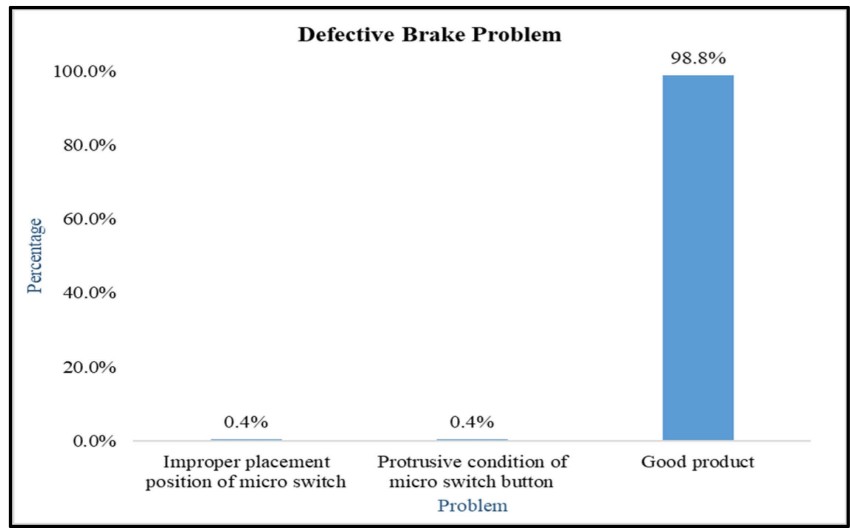

**Figure 14.** Defective brake problem.

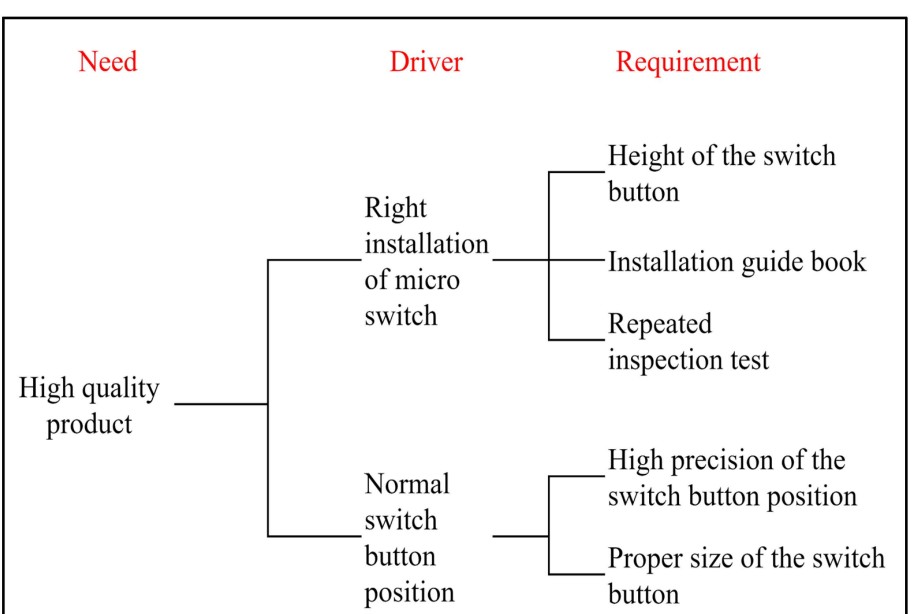

**Figure 15.** Critical to Quality (CTQ) tree of brake.

After knowing the critical problems identified on the CTQ diagram, it is necessary to conduct an in-depth analysis of at which stage of assembly the defect occurred. Therefore, the SIPOC diagram was used in this study.

The SIPOC diagram is a form used to emphasize the task through to the project's goals, and in some cases, in the detailed steps on how those goals can be accomplished. Contrastingly, SIPOC provides an overview of a project at a glance. In this study, the SIPOC diagram describes the general assembly process of model SCC4M2436A2CR6BA. Then, the output of the model is delivered to Sunrise, belonging to the Netherlands. Most of the suppliers in MTM are Taiwan companies that are loyal suppliers of MTM.

According to the SIPOC diagram in Figure 16, there are no critical problems when viewing the product assembly process sequence. However, if examined more clearly, brake

assembly is a step that needs to be considered. It happened because the brake is one of the products with defects and customer complaints.

| Suppliers | Inputs | Process | Outputs | Customers |
|---|---|---|---|---|
| Jianxin / Fu Mao | Iron / Iron Oxide | **Paste Frame Magnet** | S C C 4 M 2 4 3 6 A 2 C R G B A | SunRise (Netherlands) |
| Mengye / Jihui | PVC + Copper / Aluminium | **Bracket Wire Assembly** | | |
| Sichuan Electric / Ming Yu / Xiezhen | Iron / Iron / Copper | **Rotor Assembly** | | |
| Dongpei | Iron | **Bearing Press-in** | | |
| | | **Frame Assembly (1+2+3+4)** | | |
| Jihui | Aluminium | **Whole Unit Assembly (1+2+3+4+5)** | | |
| Fuyong | Carbon | **Carbon Brush Assembly** | | |
| | | **Magnetizing Operation** | | |
| | | **Motor Test** | | |
| Shengyu | Iron | **Brake Assembly** | | |
| Liyuan | Alumunium + Iron | **Reducer Assembly** | | |
| | | **Noise Audiometry** | | |

**Figure 16.** SIPOC DIAGRAM.

*3.3. Measure*

The measuring stage focuses on determining the problem to be a solution. At this stage, the authors are using an axiomatic design approach to overcome the design problem of the product. The axiomatic design systematically transforms the customer needs of the product into functional requirements, design processes, and process values. This approach aims to obtain the best solution to make the designer more creative, minimize the random design process, reduce the trial-and-error method, and obtain the best design within the proposed target.

Axiomatic design is taken to improve the defective product with some customer complaints. Therefore, only three domains are used in this research, the customer domain, functional domain, and design parameter.

Table 3 describes the conversion process from customer needs into functional requirements and design parameters. The conversion process is referred to Figure 8. Then, In identifying the relation between the Functional Requirement (*FR*) and Design Parameter (*DP*), the matrix in Equation (7) is used to answer the correlation between two domains.

**Table 3.** Axiomatic design of defective brake.

| Customer Need (CN) | | Functional Requirement (FR) | | Design Parameter (DP) |
|---|---|---|---|---|
| The microswitch is installed well. | $FR_1$ | Place the microswitch in the correct position | $DP_1$ | High precision of microswitch placement |
| The microswitch button is assembled well. | $FR_2$ | Assembly microswitch button properly | $DP_2$ | Repeated pressing button test |

The matrix in Equation (7) shows a full matrix in which all *FR* and *DP* relate. Each FR only fulfills one *DP*. $FR_1$ correlates with $DP_1$. High-placement precision must be corrected to make the switch fit properly. Then, a repeated button test is needed in the assembly process of the microswitch button because the button is protrusive. In addition, this kind of matrix is called an uncouple matrix, which means the design is already robust and can be applied as a new, improved design.

$$\begin{pmatrix} FR_1 \\ FR_2 \end{pmatrix} = \begin{pmatrix} X & 0 \\ 0 & X \end{pmatrix} \begin{pmatrix} DP_1 \\ DP_2 \end{pmatrix} \tag{7}$$

*3.4. Design*

The design stage of DFSS operates the execution of the measurement results. Quality control is the central aspect that needs to be considered for the company due to some defective products being one of the customer complaints. Therefore, an autonomous system must be applied to minimize human error related to product design.

To implement the design concept, an intelligent concept is applied to this stage to optimize efficiency. The design stage that must be used are as follows:

1. To develop a high infrastructure company that supports quality system concepts, upgrading and purchasing machinery and equipment are necessary;
2. To obtain high precision of microswitch placement, supportive machinery is needed;
3. A repeated button test is implied by using an innovative inspection process;
4. Determine the critical parameter of each design and assembly process to support the control process;
5. Provide the standard control of the designing and assembling process;
6. Optimize the use of machines efficiently to minimize the error (human error).

To achieve an intelligent design application for enhancing the quality aspect of the product, the above tasks must be appropriately applied. Those tasks are focused on supporting the measured result intelligently, for minimizing the defective brake. The main problem with faulty brakes is caused by human error and the wrong supplier selection process. Therefore, intelligent quality decisions must be applied to optimize time and cost through the design solution.

*3.5. Optimize*

Design cannot be perfect the first time. Therefore, an optimization process is needed to optimize the design to be more robust and resilient, so that the error design can be minimized effectively. According to the previous stage, here are the optimization tasks as follows:

1. Apply the Design Failure Modes and Effect Analysis (DFMEA) to prevent the error in advance;
2. Optimize the design process system to develop a robust design;
3. Increase the parameters of the quality control test to track the control process;
4. Optimize the precision and accuracy of machine equipment to minimize error, especially in the assembly process;
5. Implement the maintenance of machinery and equipment to prevent the error of tools;
6. Evaluate and ensure the customer needs based on the CTQ (Figure 15).

*3.6. Verify*

This final stage of DFSS is used to prove the realization process from the previous stage. Verification is necessary to minimize the error and defective products in advance. In addition, this stage is used as preventive action, regarding the unexpected problem in the future. Hence the product and process errors can be prepared from the beginning. The verification tasks of this are:

1.  Establish a quality control system supported autonomously to verify the operation and design;
2.  Pursue an automatic design process system so that it can support zero defects and the product quality can meet the customer's need;
3.  Do the routine check of the inspection process through the quality control.

**4. Discussion**

*4.1. Implementation of Six Sigma*

Six-Sigma was already implemented in 2019. MTM has experienced revenue growth, from the US $7.3 million in 2012 to US $22 million in 2020. MTM has always had "the fastest service efficiency to reach the highest customer satisfaction. The highest philosophy goal of MTM is in providing the best products and services and connecting customers to products with a competitive advantage. MTM Quality Management Systems (QMS) has the achievement of ISO-9001 certification and other patent certifications, such as ISO-14001, CE, REACH, RoHS, and UL. Therefore, MTM tried to apply for the Excellence Management Quality Award in 2020, and the Excellence Management Quality Award in 2021, and successfully won. Those achievements are extraordinary for the company. However, the implementation of Six Sigma is not very successful. This is due to a large number of defective products. Only 15% of the complaints decreased from 33% of the existing complaints. Therefore, the company needs to apply appropriate methods in dealing with current complaints to get to less than 15% of the defect number.

*4.2. Implementation of DFSS*

The number of complaints obtained by the company is still high, so the company needs to consider the problem seriously; it can increase the failure cost of the company. The data we have to emphasize the failure cost problem are insufficient because the company started this concept in 2020. Therefore, we made a linear relationship where the increase in the number of defects from customers will increase the failure cost. That is why in the study case, we focus on the analysis for the DFSS Project 1 to analyze the service items in MTM (see Table 4) and to reduce the number of complaints (see Table 5). This is the limitation of this single case study. Implementing DFSS is the right choice by MTM to solve these problems, namely reducing the number of customer complaints which will also affect the failure cost.

The implementation of DFSS is to redesign and resolve some design problems of the MTM's product. The design problem is not currently focusing on the product's shape but on how the assembly system processes in the assembly line.

According to the DFSS stage already completed, it finds several solutions that can be applied to the company as the new strategy. The strategy primarily focuses on the maintenance and quality system in MTM that needs extra attention to reduce the defective product. These significant business successes have encouraged MTM to implement the new DFSS model even more comprehensively. To achieve this goal for 2022, MTM has made a corporate strategic plan shown in Figure 17. Therefore, the below strategy defines how to implement the DFFS result into MTM directly.

After determining the corporate strategic plan for 2022, MTM needs to focus on the problems from each service item within the company to reach its vision. In Table 4, the complex issues for each service item are assessed, based on the current strategy and given a new recommendation to implement in 2022 using the new process and methodology of DFSS.

**Table 4.** The focus of DFSS is based on the analysis of service items in MTM.

| Service Item | Problem | Currents Strategy (Six Sigma) | Recommended Strategy (DFSS) |
|---|---|---|---|
| Supplier | Improper supplier | The supplier is selected by using the traditional supplier selection method. | Implement the AHP method for selecting the best supplier to minimize the cost, detect the profit, and save the time |
| | | Price, location, and time are the primary consideration of the company | Add new variables as the primary consideration to selecting the supplier, such as price, time, location, quality, and design. |
| Quality Control | Inspection process | An inspection test is used to check the function of the product | Develop high control system in the assembly line by implementing a machine interface |
| | | Manual test inspection by the operator in each flow process | Install sensors with Radio Frequency Identification (RFID) tags for each assembly part to implement the Smart Quality procedure with IoT System in MTM |
| Research and Development | Defective product | The traditional process is applied in designing the product and system | Establish empirical design methods such as Quality Function and Development (QFD) to identify the design and function of the product |
| Marketing | Product sales | Use the SIPOC diagram to improve product sales of the product. Then, identify the strength and weaknesses of the product and also the company. | Apply the Failure Mode and Effect Analysis (FMEA) approach to identify the product and improve the sales of the company by analyzing the failure and error of the sales product |
| | Customer feedback | Use customer feedback to identify the complaint of the customer. However, follow-up is seldom carried out. | Follow up intensively with the customer regarding the customer satisfaction and feedback form to improve the innovation of the product through Voice of Customer (VOC) |

**Table 5.** Objectives and measurements of DFSS projects.

| Design for Six Sigma Projects | Objective | Measure | Target | Process |
|---|---|---|---|---|
| Project 1 | Reduce the number of customer complaints | Number of complaints each week | Reduce < 33% | An assembly line |

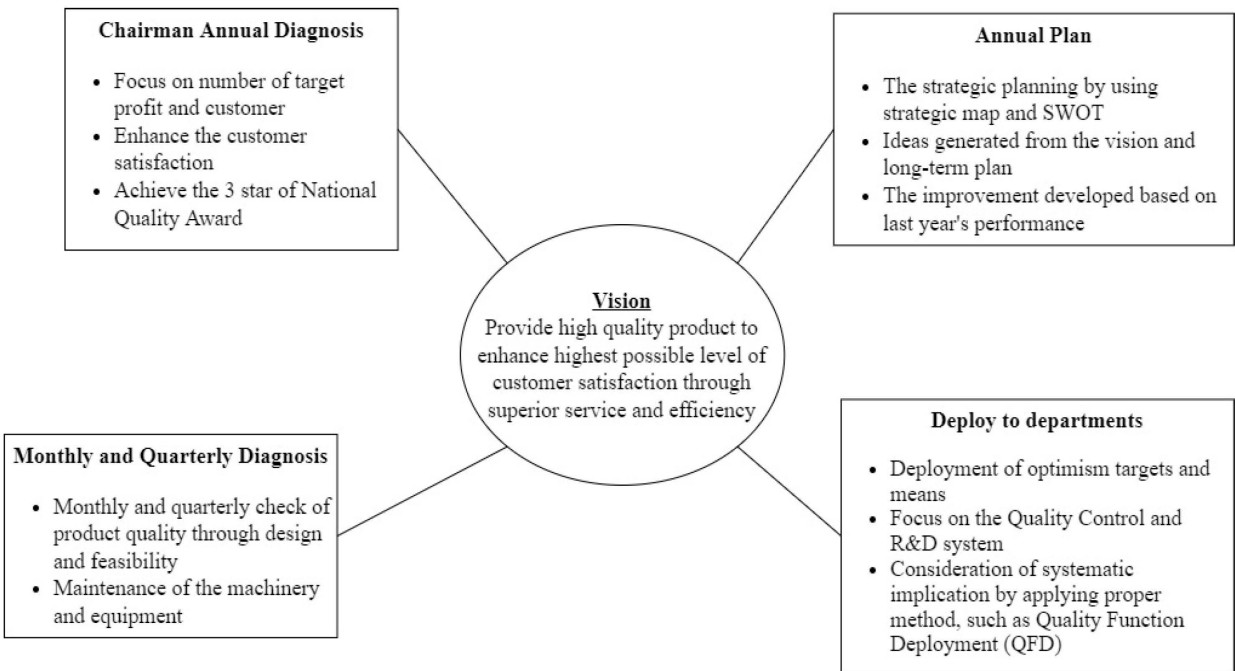

**Figure 17.** DFSS Corporate Strategic Planning of MTM.

MTM shall minimize the number of defective products by improving the quality control process. Customers' complaints from MTM prove that the quality control department must improve its performance by properly implementing intelligent quality control. In addition, the incompatibility of parts of the product with product designs designed in the previous year is another problem that causes the product to become NG.

The discrepancy was caused by not being precise in selecting suppliers for the product inputs used by MTM. Further research is needed on the selection of a more appropriate supplier. This is because it can reduce defects in the assembled product. The microswitch's wrong side has proven the supplier selection inaccuracy. To overcome this problem, MTM has identified specific DFSS projects, as shown in Table 5, and more detail described in the focus of the new development process of DFSS, based on the analysis of VOC and customer feedback for product brake SCC4M2436A2CR6BA, shown in Table 6.

**Table 6.** The focus of the new development process of DFSS is based on the analysis of VOC and customer feedback for product brake SCC4M2436A2CR6BA.

| Product Number | Product Item | Problem Identification | Current Strategy | Recommended Strategy |
|---|---|---|---|---|
| SCC4M2436A 2CR6BA | Microswitch | Wrong installation of microswitch | The design process is based on the customer feedback | Design the right height and need for the microswitch |
| | | | Improper assembly process by the operator | Implicate the standard assembly process by using autonomous techniques (robotic hand) to assembly |
| | | | Half of the Inspection process | Improve the number of inspection lines in the last assembly process. |
| | | A protrusive microswitch button | The switch button is not fit | Design the proper size of the switch button |
| | | | Manual position process | Install the switch button in the correct precision position. A sensor detector must be implemented. |

Contrastingly, the low level of inspection the company applies makes many NG products reach customers. Therefore, companies must set better standards in the final checking of products. The more NG products that match the market, the less customer trust is obtained. This will also increase the company's own failure cost. Based on the implementation concept of DFSS, MTM receive 13% (80 out of 632 pieces in a batch) of customer complaints. This number is relatively low compared to the initial 15% (75 out of 512 pieces in a batch) percentage of customer complaints. The percentage is obtained from the comparison between the number of products in production within a batch and the number of defective products. Therefore, DFSS is a suitable approach to overcome the defective product in the MTM company.

Then, to overcome the more severe problem of MTM in the future; the Quality control and Research and Development (R&D) department must be aware of routinely obtaining and collecting the production, quality control, and customer feedback data. Hence, the problem's solution can be analyzed easily without any difficulties.

Implementing Design for Six Sigma in MTM improves the following metrics and aligns them with the organization's strategic objectives. The score was collected based on the expert and management staff of the MTM company. The auditor of the MTM company rated the score of the current method MTM between Six Sigma and Design for Six Sigma in year 2020 and 2021. The managerial implications of implementing Design for Six Sigma in MTM are in Table 7 based on scores 1–10.

**Table 7.** Critical Success Factor Improvement.

| Critical Success Factor | 2020 (Six Sigma) | 2021 (Design for Six Sigma) | The Percentage Improvement |
|---|---|---|---|
| Operational efficiencies | 7.5 | 8.5 | 18% |
| Cost optimization | 7 | 9 | 36% |
| Revenue | 8 | 9 | 18% |
| Product compliance | 7.5 | 8 | 9% |
| On-time deliveries | 7.5 | 8.5 | 18% |

## 5. Conclusions

This paper looked at the design for six Sigma as an improvement strategy; it outlined the new DFSS method and the high-tech manufacturing process. It outlines how a new approach of DFSS can be applied to the high-technology company to enhance new products and manufacturing processes. Customer requirements determine the process parameters and quality indicators for high-tech products. A new DFSS process is applied to solve and improve the quality of the product through the Define, Identify, Measure, Design, Optimize, and Verify (DIMDOV). Although the purpose of this paper was to provide the applicability of the new DFSS process, it also outlined the framework for implementing the technique in the high-technology manufacturing process; the outcomes of the applied tools in the define stage and understanding the customer requirement during the measure phase. The design step aims to reduce the later product life cycle problems, especially in manufacturing. The robust design and optimal control of the manufacturing process can be achieved in the optimized step. The verify step approves the optimal design works in the previous actions. Moreover, for the high-technology manufacturing company, it becomes relevant due to the production requirement of the technology. The high-technology company requires process design optimization every time a new product is manufactured.

The revenue and growth of most companies are directly related to how high-technology companies, including MTM, successfully launch new products and improve their processes. A new approach of DFSS with DIMDOV steps can serve as a mechanism to revolutionize how high-tech companies develop new products. To meet and satisfy customer requirements, the DIMDOV framework is a practical approach that can be used to ensure that products meet specifications. This framework covers all aspects of product design, including product definition, critical requirement definition, design optimization, and quality. With tools such as Kano analysis, refined Kano, QFD, axiomatic design, DFMEA, DOE, Taguchi, simulation, and reliability analysis, the new process of DFSS has proven to be an effective method for developing new high-tech products. The use of DFSS in this study has dramatic impact, reducing the number of complaints. To ensure the effectiveness of the MTM improvement project, managers at all levels must commit to investing resources to initiate, promote, actualize, and support the program. Hence, the company's target to reduce 15% of customer complaints in a year can be handled perfectly.

In addition, there are some limitations of a new approach to DFSS. Consolidating and modernizing the legacy quality control and assessment system requires an extra cost. The cost is used to provide appropriate equipment and machinery. Therefore, the company must spend expenses and time. This transition also needs enough training for the member. Hence, the implementation of DFSS can run well. This kind of aspect can be considered the weakness of DFSS for the company. Innovation is essential to implement, but the cost is the main central aspect.

Future work will involve the application of the new DFSS to another type of industry and recording its outcomes. A case study methodology should be used to implement the outlined new DFSS methodology and to study the consequence effect of applying the technique. Contrastingly, developing intelligent manufacturing is a long-term and challenging journey. It may take more than five years. Therefore, the development and design stage are critical in developing innovative quality control systems in the future study of this research.

**Author Contributions:** Y.-T.J. contributes in conceptualization, methodology, supervision, validation, and writing down the original draft. C.-C.Y. takes a part as project administration, resources, and supervision. M.-C.L. as a third author involves in funding acquisition, investigation, resources, and writing the review, and editing. Then, R.M.S. contributes as a formal analyst, investigator, graphic designer, and also as a editor. The last author R.S. involves in investigation, visualization, and editor. All authors have read and agreed to the published version of the manuscript.

**Funding:** This paper is funded by the author itself without any external support.

**Data Availability Statement:** This study focuses on qualitative research by using interview data result from the MTM company. The company's data is confidential. Therefore, it cannot be put in any open public platform. On the other hands, the profile of this company can be seen in this link https://motiontech.en.taiwantrade.com/.

**Acknowledgments:** The authors would like to thank Motion Technology Electric and Machinery, Co., Ltd. (MTM), Taoyuan City, Taiwan (R.O.C.) as the subject for a case study, for their permission to collect the data. They would also like to thank to the anonymous reviewers for their useful suggestion.

**Conflicts of Interest:** The authors declare no conflict of interest.

## Appendix A

The Appendix A is the interview result related to the customer complaint and problem in MTM company.

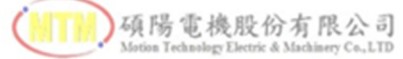

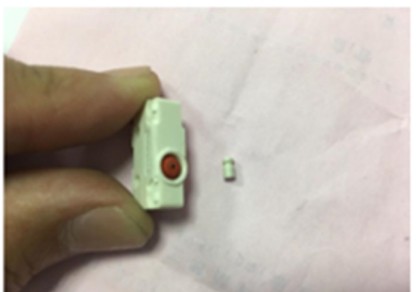
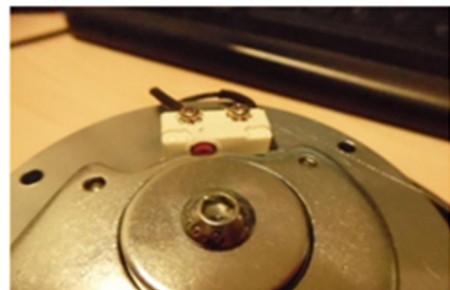

Defect problem

---

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
