# Peer review of "The Development of the New Process of Design for Six Sigma (DFSS) and Its Application"

_sustainability, doi:10.3390/su14159294_

Round 1

Reviewer 1 Report

The paper should have the following shortcomings. The writing needs improvement. The advance of the proposed method needs to be verified by the data of actual cases. Detailed suggestions are as follows.

1.      In the introduction, please introduce the current application status of DFSS.

2.      What are the disadvantages of DFSS?

3.      What is the innovation of this paper?

4.      Figure 1 is too vague.

5.      Lines 453-456, Lines 465-468, two repeated expressions.

6.      Line 479, it should be clearly stated the number of Six Sigma projects.

7.      Line 480, “Information for was obtained through a series of structured interviews with Six 480 Sigma project team leaders, and members;” should be rewritten.

8.      Line 482, The questionnaire can be attached as the appendix.

9.      Line 498, figure 12, or Figure 12?

10.   Figure 13 doesn’t clearly explain what’s the defective. Please mark the defectives on the pictures.

11.   Line517,Figure 13 should appear after the text that mentions it in the text.

12.   Lines 539-559, what is the SIPOC? what is the relationship between CTQ and SIPOC?

13.   Line 663-664, what’s the basis for this conclusion?

14.   Line 655, 15% of the number of complaints decreased from 33% of the existing complaints. Line 702, Based on the implementation concept of DFSS, MTM can get 15% of customer complaints. 702. 15%. The 15% complaint rate was obtained after application of 6sigma or DFSS?

15.   In this paper, DFSS and 6sigma are compared, but the quantitative comparison of the improvement effect of the two methods is lacking. Has DFSS not completed the whole process in the company? If so, how can DFSS be validated?

16.   At the same time of quality improvement, comparative analysis of improvement cost and improvement benefit is needed.

17.   How advanced is DIMDOV compared to conventional DFSS represented in the case?

18.   The author has a wealth of experience, why not do statistical analysis of all the cases?

Author Response

Dear Reviewer 1,

Thank you for giving us the opportunity to submit a revised draft of our manuscript titled “THE DEVELOPMENT OF THE NEW PROCESS OF DESIGN FOR SIX SIGMA (DFSS) AND ITS APPLICATION” to the Sustainability Journal. We appreciate the time and effort that you and the reviewers have dedicated to providing your valuable feedback on our manuscript. We are grateful to the reviewers for their insightful comments on our paper. We have been able to incorporate changes to reflect most of the suggestions provided by the reviewers. We have highlighted the changes within the manuscript.

The point-by-point response to the reviewer’s comments and concerns is attached.

Thank you for allowing us to submit a revised draft of our manuscript. We look forward to hearing from you regarding our submission and responding to any further questions and comments you may have.

July 11, 2022

 Sincerely,

Ronald Sukwadi

Reviewer 2 Report

The paper suits the scope of the journal and Sustainability and its special issue.

The structure of the paper follows a structure of a scientific paper.

My comments for improvement of the paper are as follows:

1.   Paper´s aim: state it clearly from the scientific point of view in abstract and introduction.

2.  Introduction: add information about the main contribution of the paper and better underline the novelty of the pap

3. Material and methods: improve the layout of figures (smaller font, less colours). DFSS model described is adopted or created by authors? – please make it clear.

4. Discussion of results with other works is missing - it should be added.

5. Conclusions - add information:

-  Highlight the contribution of the paper to development of scientific knowledge

-  Give limitation of the study/research presented in the paper

Author Response

Dear Reviewer 2,

Thank you for giving us the opportunity to submit a revised draft of our manuscript titled “THE DEVELOPMENT OF THE NEW PROCESS OF DESIGN FOR SIX SIGMA (DFSS) AND ITS APPLICATION” to the Sustainability Journal. We appreciate the time and effort that you and the reviewers have dedicated to providing your valuable feedback on our manuscript. We are grateful to the reviewers for their insightful comments on our paper. We have been able to incorporate changes to reflect most of the suggestions provided by the reviewers. We have highlighted the changes within the manuscript.

The point-by-point response to the reviewer’s comments and concerns is attached.

Thank you for allowing us to submit a revised draft of our manuscript. We look forward to hearing from you regarding our submission and responding to any further questions and comments you may have.

July 11, 2022

 Sincerely,

Ronald Sukwadi

Reviewer 3 Report

The overall design goal of this paper is achieved. The topic of this paper has certain research significance and practical application value, and can solve the specific functional requirements in practical application.

The author of this paper can combine the actual needs, conduct basic research on the content of this research field, basically understand the work of predecessors in this field, and the background of the subject is clear.

The paper has certain technical difficulties, and the workload is basically full.

Author Response

Dear Reviewer 3,

Thank you for giving us the opportunity to submit a revised draft of our manuscript titled “THE DEVELOPMENT OF THE NEW PROCESS OF DESIGN FOR SIX SIGMA (DFSS) AND ITS APPLICATION” to the Sustainability Journal. We appreciate the time and effort that you and the reviewers have dedicated to providing your valuable feedback on our manuscript. We are grateful to the reviewers for their insightful comments on our paper. We have been able to incorporate changes to reflect most of the suggestions provided by the reviewers. We have highlighted the changes within the manuscript.

The point-by-point response to the reviewer’s comments and concerns is attached.

Thank you for allowing us to submit a revised draft of our manuscript. We look forward to hearing from you regarding our submission and responding to any further questions and comments you may have.

July 11, 2022

 Sincerely,

Ronald Sukwadi

Round 2

Reviewer 1 Report

Whether the new theory and method are effective, what are the advantages, and what are the shortcomings, all these need to be verified based on practice and tests.

However, the main shortcoming of this paper is that the validity of the proposed method has not been verified by a complete application cycle.

I suggest that the authors should conduct quantitative analysis based on actual data to verify the effectiveness of the proposed method and clarify its advantages and disadvantages, after completing several rounds of DIMDOV application in MTM and other companies.

At the same time, the cost and benefit of the improvement method should be analyzed, and the results should be quantitatively compared with the original improvement method implemented by the company.

In addition, for DMADV, why is the “A” phase removed? For DIMDOV, although the reason for adding “I” phase has been explained,  the reason for adding “O” phase should also be explained.

Author Response

Thank you for giving us the opportunity to submit our second revised draft of our manuscript titled “THE DEVELOPMENT OF THE NEW PROCESS OF DESIGN FOR SIX SIGMA (DFSS) AND ITS APPLICATION” to the Sustainability Journal. We have highlighted the changes within the manuscript. Here is a point-by-point response to the reviewer’s comments and concerns.

Round 3

Reviewer 1 Report

The experts and managers involved in the scoring need to be analyzed and explained to ensure the effectiveness of the scoring.

Based on the implementation concept of DFSS, MTM can get 13% of customer complaints. Does the word "can" mean that customer complaint rates have not been achieved through practice? If so, how was the 13% customer complaint rate obtained? If the data were based on analysis, please explain the analysis process.
